# Measurement and evaluation of low-carbon tourism development on islands: A case study in Changdao, China

**Mengsha Wang**[1]☯, **Jiayu Zuo**[2]☯*

1 School of Economy, Shandong Technology and Business University, Yantai, Shandong, China,
2 Business School, Hanyang University, Seoul, Korea

☯ These authors contributed equally to this work.
* jyzuo5767@sina.cn

**Data Availability Statement:** The data are available from figshare at https://doi.org/10.6084/m9.figshare.28006280.v1. The data are from China Environmental Monitoring Centre (http://www.

## Abstract

China's island tourism is still in the exploratory stage, and the carbon emissions due to island tourism development are still prominent. This study assesses the development of low-carbon tourism on Changdao Island in China. We constructed an evaluation model for low-carbon tourism on islands based on the driver-pressure-state-impact-response model, and the Entropy Weight Method-Analytical Hierarchy Process Method was combined with the weighting method to determine the index weights of ench evaluation-indicator. The annual changes in the development level of low-carbon tourism, the weights of the indicators, the characteristics of the scores, the low-carbon development mode, and key factors of island tourism were analyzed. The results of the study showed that the indicator "impact" was the most influential element of the island's low-carbon tourism, and the ecological environment value was higher than that of the economic value. Moreover, energy saving and consumption reduction helped tourists to have a better experience, which further enabled island tourism to have a larger market size. The "response" also occupied a crucial position, where the weighted value of government planning was twice as high as that of corporate practice. The other elements were "pressure", "state", and "driving forces", which showed that low carbon emissions are an important criterion for the island tourism environment and economic factors have the greatest effect in terms of the "driving forces". The study evaluated the level of low-carbon development in island regionals across multiple dimensions, filled the literature gap, and provided a reference for the study of regional low-carbon and sustainable development of tourism.

## Introduction

United Nations released the Implementation Schedule for the Ocean Decade", which strives to attain long-term sustainability of the ocean by establishing a Common Framework of Global Maritime Interests [1,2]. According to the EU Blue Economic Report 2022 that was issued by the European Commission, the total added value of the blue economy industry reached 184 billion euros in 2022, but rising sea levels may cause huge losses in the future [3]. Moreover, China's "Outline of the 14th Five Year Plan and the Long-term Objectives for 2035"

cnemc.cn), Shandong Ecological Environment Monitoring Centre (http://wap.sdein.gov.cn), and Ctrip Group Limited's Ctrip.com (this study used the template in Octopus Collector to collect the hotel and review information from Ctrip.com). Researchers who meet the criteria for confidential data access can obtain data from The Marine Ecological Civilization Comprehensive Experimental Area of Changdao Institutional Data Access (https://www.changdao.gov.cn/col/col15722/index. html).

**Funding:** This study was financially supported by Shandong Technology and Business University's Doctoral Introduction Start-up Fund Project "Research on Economic Benefits and Development Countermeasures of Carbon Sink Fisheries in the Blue Economic Zone of Shandong Peninsula" [BS2021144] to MW. The funders had no role in study design, data collection and analysis, decision to publish, or preparation of the manuscript.

**Competing interests:** The authors declare there is no conflict of interest.

emphasized the revitalization of the rural economy and industries by developing characteristic industries such as rural tourism and homestays, the promotion of a modern marine system, and the development of marine power [4]. Consequently, the development of ocean tourism in coastal areas and islands aligns with the strategic goals of rural revitalization and the development of "maritime power".

During 2012–2021, the added value of the Marine tourism industry increased from 33.9% to 44.9%, and it has become a pillar industry for the ocean economy and playing an important role in boosting consumption, domestic demand, and economic growth. Marine tourism has a long industrial chain, which involves the six traditional elements of food, accommodation, transportation, tourism, shopping, and entertainment and the modern development elements of experience, meeting, health, media, organization, and supporting facilities. It also has a strong correlation with upstream and downstream industries and plays a critical role in promoting regional, national, and global economic growth. Moreover, as the offshore fishery resources deplete, island tourism is becoming a powerful engine for rural revitalization, economic growth, and demand expansion in islands.

The rapid economic progress in China's coastal areas has led to a sequence of marine environmental problems, such as the increase in marine pollutant emissions and the deterioration of marine ecology in the coastal waters [5]. As an important part of the marine ecosystem, islands are where many plants and animals live, reproduce, and stopover during migration and are ecologically sensitive areas. Therefore, their ecology is more vulnerable to various environmental problems due to human activities. However, although tourism is considered a "green and smoke-free industry", excessive and uncontrolled tourism and development lead to expanded carbon footprints, and the consequence of carbon emissions from tourism on the ecological environment cannot be ignored [6–8]. According to the World Tourism Organization Report, tourism has become a "major contributor" to the growth of carbon emissions and its carbon emissions continue to grow.

The study of energy consumption and carbon emissions within the tourism sector has gained significant interest among scholars in recent years. Since the 1990s, research on energy consumption in the tourism industry, such as that conducted in the Hawaiian Islands [9], has become a growing area of interest. Various approaches have since been employed to assess carbon emissions in tourism, including top-down method [10–12], bottom-up method [13–15], Tourism Satellite Account estimates [16,17], and Life Cycle Assessment [18]. The findings suggest that the tourist scale effect and energy intensity contribute to increasing carbon emissions [19]. Furthermore, carbon intensity, income structure, consumption level, and tourism scale have been identified as four key factors affecting tourism carbon emissions [20]. While most of the literature has focused on the estimating carbon emissions, few studies have measured the overall green and low-carbon level of island tourism and provided possible measures to enhance zero-carbon development, we attempted to explore these aspects.

This study created a low-carbon island tourism development level evaluation index system to measure and evaluate the development level of low-carbon island tourism and explore how different elements in the island tourism system are integrated and developed. This study aimed to provide a theoretical basis for the green, low-carbon, and sustainable development goals of island tourism, contribute to the healthy economic and social development of islands, and promote low-carbon tourism for the revitalization of fishing villages on islands. The study could be used as a basis for contributing toward achieving sustainability in the global ocean economy and construction of international zero-carbon islands in China. Moreover, island tourism destinations are relatively independent geographical units and special tourism destinations, and thus exploring their low-carbon development modes for scientific policy control,

construction planning and the construction of "zero-carbon" demonstration zones can provide a basis for the construction of zero-carbon communities.

## Literature review

### Low-carbon tourism

With the promotion of global carbon reduction, carbon emissions from the tourism industry have attracted attention from countries around the world, and related theories and policies have surged. Hence, increasing research has been conducted on low-carbon tourism as an emerging tourism paradigm. Low-carbon tourism is a sustainable development approach for the tourism industry that utilizes low-carbon technologies and management methods, intending to reduce carbon emissions and improve tourism quality through measures to reduce carbon emissions and pollution from transportation, accommodation, sightseeing, shopping, and entertainment [21]. The development of low-carbon tourism can result in economic, social, and environmental benefits to tourist destinations, cities, and regions, and is a common choice for promoting tourism and economic development [22,23]. There are interrelationships among the economy, society, and environment of tourist destinations. Additionally, sustainable development is a multidimensional concept that includes the economy, society, and environment [24]. Thus, tourist destinations contain many interdependent and highly nonlinear relationships, manifested as dynamic and constantly evolving complex systems [25]. However, research on low-carbon tourism has mainly focused on single aspects such as low-carbon tourism products or activity planning, low-carbon tourism behavior analysis based on tourist perspectives, and evaluation of carbon emissions from tourism [26–29].

Zhang and Zhang [30] examined tourist destination cities and constructed an evaluation index system encompassing three dimensions: low-carbon tourism economy, low-carbon tourism environment, and low-carbon tourism society. Their study highlighted the importance of carbon dioxide emissions and energy consumption in hotels, tourist attractions, and other tourism enterprises, as key economic and environmental indicators. They also emphasized the low-carbon literacy of tourism stakeholders, including residents, tourists, and tourism enterprises. Fakfare and Wattanacharoensil [31] analyzed low-carbon tourist behavior on island destinations by integrating environmental inputs, emotional states, and behavioral responses. They identified key influencing factors, such as service providers, tourism activities, and environmental management. These studies offer theoretical support for the constructing a comprehensive and scientific low-carbon tourism evaluation system based on a system perspective and the characteristics of island destinations in this study.

### Sustainable development of island tourism

The sustainable development of tourist destinations involves responsible management that maximizes their potential while balancing important economic, environmental, and socio-cultural factors [32]. Islands have long been popular tourist destinations due to their unique resources. However, the fragile nature of small island environments has led to increased research on the sustainable development of island tourism. Given the relative isolation of islands, it is possible to disentangle the causal effects of island tourism development on their society, environment, and economy and conduct focused research on sustainable island tourism development [33]. Promoting sustainable development in island regions and island tourism requires careful management of stakeholder relationships (among tourism enterprises, tourists, the tourism service industry, residents, and the government) to effectively manage public resources and foster collaboration [34,35]. Therefore, to address the complex challenges of sustainable island tourism, a dynamic tourism system is needed that includes all subsystems

(such as economic, social, cultural, ecological, technological and political dimensions). These systems must be evaluated and optimized to establish effective management systems and guiding policies [36–38].

Most of the empirical studies on the evaluation of island tourism development in China have analysed the coupling coordinated development level of the island tourism systems by constructing an evaluation index system for all or two of the three factors, namely, tourism, the socio-economy, and the ecological environment. The results of these studies showed that the level of coupling and coordination between island tourism and the ecological environment [39,40] economic growth [41], and socio-economics [42] in China is not high but is gradually increasing. The tourism system is a dynamic system that is composed of multiple elements, and it requires the organic integration and coordination of various aspects. Constructing this system is an important step in the transition from theoretical research to practical application, and it is an effective way to achieve economic, social, and ecological environmental benefits. It is also a key part of relevant policy formulation and construction planning management [43–45].

## Driving force-pressure-state-impact-response framework

Studies on the low-carbon aspects using the driving force-pressure-state-impact-response (DPSIR) model have focused on low-carbon cities and transportation. Integrating the DPSIR model into low-carbon cities and transportation management allows for the incorporation of diverse index dimensions, facilitating the comprehensive consideration of both direct and indirect impacts. This approach helps establish a reliable evaluation index system that enables policymakers to evaluate and monitor environmental trends effectively [46,47]. The DPSIR model combines the strengths of the pressure-state-response (PSR) and driving force-state-response (DSR) models, considering five major factors (economic, social, resource, environmental, and policy). As a result, it offers a comprehensive, systematic, adaptable, and holistic framework. Due to these qualities, the DPSIR model is widely applied in sustainable development and low-carbon research [43,48,49].

Then studies on island systems using the DPSIR model have focused on the two aspects of ecological safety and sustainable development [50,51]. Research on tourism systems based on the DPSIR model has mainly focused on ecological security assessment and sustainable tourism assessment [52,53]. For example, Ruan et al. [54] proposed that constructing an evaluation model can be used to explore the current situation and evaluation index system through quantitative and qualitative analysis, and the DPSIR model comprehensively examines the interrelationships between people and the environment from a systematic perspective. Thus, it fully reflects the interaction between tourists, tourist destinations, and the environment and the continuous feedback mechanism between monitoring indicators.

A review of relevant literature revealed that research on low-carbon tourism on islands has mostly focused on the perspective of tourists, exploring their low-carbon behavior preferences, including low-carbon attractions, low-carbon transportation facilities, environmental protection products, low-carbon projects, low-carbon hotels, low-carbon catering, and shopping activities [31]. However, the development of island tourism and the level of low-carbon development depend on a wide range of stakeholders and local economic, social, cultural, and environmental factors. The research has addressed low-carbon tourism, the transition to low-carbon destinations, and the evaluation of low-carbon competitiveness. Some studies have used the DPSIR model and have accounted for the carbon emissions of the tourism industry or specific localities, developing a low-carbon development evaluation index system to address sustainability challenges [55–57]. Therefore, we plan to apply the DPSIR model to construct an accurate and scientific evaluation index system for low-carbon tourism on islands.

### Research gap

This literature review not only examined the theoretical foundations of low-carbon tourism and island tourism, but also addressed the theoretical foundations of the methodology that is used in this study. The existing research on tourism destinations and low-carbon development based on the DPSIR model provides a foundation for constructing the theoretical framework of this study. However, no research has been conducted on low-carbon island tourism development using the evaluation system based on the DPSIR model with island systems as the focus and carbon emissions as a specific index.

Consequently, this study conducted a comprehensive analysis of links between the economy, society, resources, environment, policies, and carbon emissions that were related to island tourism development from a system analysis perspective. Based on the DPSIR analysis model and with carbon emissions from tourism as the core index, this study constructed an index system that reflects the overall development level of low-carbon island tourism. The index included five sub-systems of the low-carbon tourism system, that is, its potential driving forces, direct explicit pressure, core carbon source department state, socio-economic environment impact, and government and enterprise response. The low-carbon tourism development level of Changdao Island was analyzed, which could provide theoretical guidance for building a comprehensive marine ecological civilization pilot zone and formulating low-carbon development policies for Changdao Island.

## Methods

### Research design

First, the carbon emissions of each major tourism department were calculated separately. Secondly, to comprehensively assess the development level of the tourism carbon emissions in Changdao Island, we selected five criterion layers, namely driving, pressure, state, impact, and response, according to the DPSIR model. These criterion layers corresponded to various kinds of small indicators. Then, the basic data of the various indicators in the system were collected from 2016 to 2021, and their characters were judged according to their positive or negative impacts on tourism carbon emissions. Thirdly, according to the characters and data of each year, we obtained the information entropy and weights of these indicators in the total system using the entropy weighting method. To ensure the comprehensiveness of the system, we also obtained the Analytical Hierarchy Process (AHP) weights through the AHP method based on the ratings of several experts for the different carbon emission indicators. Fourth, the entropy weights and AHP weights were combined in equal proportions to obtain the comprehensive weights for the overall carbon emission environmental system. Fifth, while the entropy weights fluctuated based on yearly statistical data, the AHP weights derived from expert scores remained consistent each year. By synthesizing the two, the weights of each indicator were calculated annually, and the scoring value of each criterion layer of the environment system was obtained for the carbon emission impact indicators of the system. These scores represented yearly comprehensive assessments of the entire low-carbon island tourism emission system.

**Tourism carbon emissions measurement.**   Previous studies on carbon emissions from different tourism departments revealed that tourism transportation, tourist accommodation, and tourism activities accounted for more than 90% of carbon emissions from tourism, indicating that they showed a long tail effect. Hence, the carbon emissions from tourism were measured according to the principle of "Occam's razor", with the carbon emissions of the three core carbon source departments as the measurement objects [24]. Consequently, the carbon

emissions from tourism were calculated as follows:

$$C = \sum_{i=1}^{n} C_i \tag{1}$$

$$C_t = \sum_{i=1}^{n} p_{ti} d_i k_{ti} \tag{2}$$

$$C_h = 365 \sum_{i=1}^{n} q_i r_i \beta_i e \tag{3}$$

$$C_a = \sum_{i=1}^{n} p_{ai} k_{ai} \tag{4}$$

where $C$ is the total carbon emissions from tourism, $C_i$ is the total carbon emissions from the different tourism departments, $C_t$ is the total carbon emissions from tourism transportation, $C_h$ is the total carbon emissions from tourist accommodations, and $C_a$ is the total carbon emissions from tourism activities; $q_i$ is the number of beds in the accommodation mode $i$, $r_i$ is the average daily occupancy rate of the beds in the accommodation mode $i$, and $\beta_i$ is the unit energy consumption of the accommodation mode $i$ (in MJ/night); $e$ is the conversion coefficient of the energy consumption and carbon emissions (in g/MJ); $p_{ai}$ is the number of tourists choosing the tourism activity $i$, and $k_{ai}$ is the carbon emissions from the tourism activity $i$ (in g/capita).

In terms of the statistics of the carbon emissions from tourism transportation, to accurately estimate the carbon emissions from tourism transportation for off-shore island tourism destinations, the "central transportation and internal transportation to the scenic spot" was specified by the narrow statistical boundary and was used as the statistical object [36]. Based on the waterway tourist turnover, road tourist turnover, and field research, the proportion of tourists that chose each transportation mode was obtained. Based on the obtained data, the tourist turnover ($p_{ti} d_i$ in Eq (2)) of each transportation mode was calculated. Then, according to the $k_{ti}$ carbon emission coefficient, the carbon emissions of each transportation mode were obtained. In terms of carbon emissions from tourism accommodation, considering that some of the fishtainment homestays were ultra-small-scale businesses that were self-owned by families and considering the availability of the statistical data, the measured carbon emissions from tourist accommodation in Changdao Island were limited to star-rated fishtainment homestays and hotels. The carbon emissions from tourist accommodation were calculated using Eq (3). In terms of the carbon emissions from tourism activities, according to the analysis of the tourist motivation based on the accommodation reviews, the proportion of tourists for each type of tourism activity and the corresponding person-times were determined.

**Low carbon tourism evaluation index system construction.** *Driver-pressure-state-impact-response model.* When compared with the PSR model and the DSP model, the DPSIR model comprehensively covers and integrates the five elements of economy, society, resource, environment, and policy. It can also give a clear, complete, and dynamic reflection of the causal, constraint, and feedback relationships of the five indexes, namely driving forces, pressure, state, impact, and response, by establishing a complete D-P-S-I-R causality chain from the perspective of system analysis. This model was initially used in sociological research, but it has gradually been used in the fields of sustainable development and ecological environment, and now it has been applied to studies to evaluate the development of cities, transportation, and other research objects in the low-carbon field.

*Construction of an evaluation system.* The tourism ecosystem covers aspects, such as economy, society, environment, and tourism development [39]. Additionally, the low-carbon development assessment system covers indicators, such as carbon emissions and carbon emission intensity

[56]. Based on relevant literature, policy documents, and reports, it is proposed that the DPSIR model framework should cover dimensions, such as tourism industry development, local economy, society, resources, ecological environment, carbon emissions status, policy guidance, and corporate practices. Hence, to investigate the relationship between tourism, socio-economic development, and carbon emissions in tourism-based island systems, an evaluation system was constructed that included an objective layer for the overall evaluation of the island tourism low-carbon level, five criterion layers, namely, driving forces, pressure, state, impact, and response, the element layers of all the criterion layers, and an indicators layer for specific measurement (Fig 1).

The driving forces drive the development of island tourism and are the potential cause for an increase or reduction in carbon emissions. Furthermore, the driving force is an important input factor for the functioning of the tourism system, and it promotes the development of the tourism industry [54,58]. The economic and social macro environment of tourist destinations provides essential support for the development of the local tourism industry. Moreover, the natural and social cultural resources and the location and climate conditions of tourist destinations serve as key attractions for the development of the tourism industry. Based on existing studies, four driving elements (economy, society, resource, and location) and 21 indexes were preliminarily determined.

The pressure is an explicit index that is directly affected by the development of island tourism under the internal action of the driving forces, and it has an impact on the carbon emissions of the islands. "Pressure" poses a significant threat to the low-carbon development of tourist destinations as various tourism activities can increase resource consumption, environmental pollution, and ecological damage. The primary sources of carbon emissions from tourism include transportation, accommodation, and activities. Therefore, four element layers (island tourism market size, island transportation, island accommodation, and island tourism activity) and 16 indexes were preliminarily determined.

The total carbon emissions and carbon intensity indicators are commonly used to quantitatively analyze the tourism footprint and evaluate the low-carbon development of tourism. This study highlights the importance of carbon dioxide emissions and energy consumption from tourism-related sectors, including transportation, accommodation, and tourism activities, in terms of their impact on the low-carbon economy and environmental indicators. The state is the carbon emission state of island tourism and its sub-sectors of transportation,

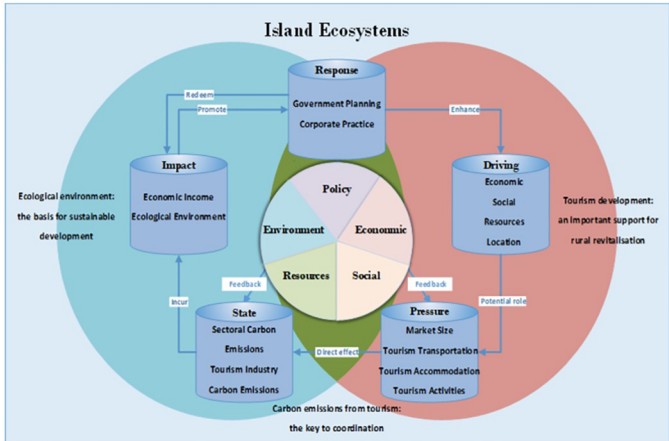

**Fig 1. Driver-pressure-state-impact-response model for the evaluation of low-carbon tourism development on islands.**

accommodation, and tourism activity under the internal driving forces and direct pressures, and two element layers (tourism and sub-sectors) and five indexes were preliminarily determined.

The impact involves the well-being of human beings in the socio-economic system of tourist destinations [52]. For island tourism destinations, the development of low-carbon tourism has contributed to increased local tourism revenue and economic conditions. Moreover, the low-carbon development of tourism positively affects the local ecological environment. Therefore, in this study, the impact is the impact on the economic income and ecological environment of the islands due to carbon emissions from island tourism, and two element layers and 14 indexes were preliminarily determined.

Response measures involve the policies and management strategies that are proposed by governments and stakeholders to consciously organize and plan low-carbon tourism to address DPSIR issues [53]. In this context, the government and tourism enterprises are considered the two key entities capable of significantly influencing the development of low-carbon tourism. Therefore, in this study, the response is the response to the economic, social, and ecological impacts on the islands, mainly in terms of the measures that were taken to promote low-carbon island tourism development, and two element layers (government [macro-control side] and enterprises [intermediary side]) and 10 indexes were preliminarily determined.

There were 14 element layers and 86 indexes in total. After the elimination of inappropriate, duplicate, and controversial indexes through expert consultation and statistical analysis and those with missing data based on data query and consultation, a total of 14 element layers and 49 indexes were finally determined in Table 1.

*Determination of Index Weights.* After the data from the low-carbon island tourism development evaluation index system were obtained, the raw data of the varying attributes and dimensions were nondimensionalized to obtain standardized and comparable data. The extreme value processing method was used to substitute the positive and negative indexes into Eqs (5) and (6) as follows:

$$X_{ij} = \frac{I_{ij} - I_{min}}{I_{max} - I_{min}} \tag{5}$$

$$X_{ij} = \frac{I_{max} - I_{ij}}{I_{max} - I_{min}} \tag{6}$$

Since the AHP method requires experts to compare the importance of one index with another index for each layer and then assign weights subjectively according to the scale table, it has the disadvantage of being influenced by subjective factors. However, the entropy weight method also has the disadvantage of purely objective weighting. Hence, this study used the Entropy Weight Method-AHP combined with the weighting method to determine the index weights, which reduced the disadvantages of both methods and had the advantages of high rationality, systematicness, and accuracy.

A judgment matrix was constructed for the criterion layers and five indicator layers based on the export scores. As seen in Eqs (7)–(10), the AHP weights of the criterion layers and indicator layers were obtained. Also, a consistency test was performed on the judgment matrix.

$$w_i^0 = \frac{\left(\prod_{j=1}^{n} a_{ij}\right)^{\frac{1}{n}}}{\sum_{i=1}^{n} \left(\prod_{j=1}^{n} a_{i,j}\right)^{\frac{1}{n}}} \ (i = 1, 2, 3\ldots, \mathrm{n}) \tag{7}$$

**Table 1. Island low carbon tourism development evaluation index system.**

| Target | Criterions | Elements | Indicators | Character | References |
|---|---|---|---|---|---|
| Comprehensive evaluation index of the level of development of low carbon tourism in the islands ($O$) | Driving ($C_1$) | Economic($E_1$) | Percentage of tertiary industry (%) $I_1$ | Positive | Ruan et al. (2019)[54] |
| | | | GDP per capita (RMB/person) $I_2$ | Positive | |
| | | | Resident disposable income per capita (RMB) $I_3$ | Positive | |
| | | | Resident consumption expenditure per capita (RMB) $I_4$ | Positive | |
| | | Social($E_2$) | Population size (People) $I_5$ | Positive | Ruan et al. (2019)[54]; Chen et al. (2022)[59] |
| | | | Natural population growth rate (‰) $I_6$ | Positive | |
| | | | Urbanization level (%) $I_7$ | Positive | |
| | | | Businesses in the accommodation and catering industry (Unit) $I_8$ | Positive | |
| | | Resources($E_3$) | National A-class scenic spots (Unit) $I_9$ | Positive | Zha et al. (2019)[34] |
| | | | Forest coverage (%) $I_{10}$ | Positive | |
| | | | Coastline length (km) $I_{11}$ | Positive | |
| | | | $CO_2$ absorption (ton) $I_{12}$ | Positive | |
| | | Location($E_4$) | Periods of suitable tourist temperatures (day) $I_{13}$ | Positive | Matzarakis (2006)[60] |
| | Pressure ($C_2$) | Market Size($E_5$) | Tourist numbers (10,000 people) $I_{14}$ | Positive | Zhang et al. (2022)[58] |
| | | | Number of people hosted on sea excursions (10,000 people) $I_{15}$ | Positive | |
| | | | Electricity consumption (10,000 kWh) $I_{16}$ | Inverse | |
| | | Tourism Transportation ($E_6$) | Road passenger traffic (10,000 visitors) $I_{17}$ | Positive | Zhang et al. (2022)[58]; Cho et al. (2015)[61] |
| | | | Marine passenger traffic (10,000 visitors) $I_{18}$ | Positive | |
| | | | Road passenger traffic turnover (10,000 visitors km) $I_{19}$ | Positive | |
| | | | Marine passenger traffic turnover (10,000 visitors km) $I_{20}$ | Positive | |
| | | Tourism Accommodation($E_7$) | Average daily star hotel bed occupancy (Unit) $I_{21}$ | Positive | Zhang and Zhang (2020) [30]; Cho et al. (2015) [61] |
| | | | Average daily star fisherman's family home hotel bed occupancy (Unit) $I_{22}$ | Positive | |
| | | Tourism Activities($E_8$) | Number of tourists on sightseeing tours (People) $I_{23}$ | Positive | Zhang and Zhang (2020) [30]; Cho et al. (2015) [61] |
| | | | Number of visitors on leisure holidays (People) $I_{24}$ | Positive | |
| | | | Number of visitors on business trips (People) $I_{25}$ | Positive | |
| | | | Number of visitors for other tourism purposes (People) $I_{26}$ | Positive | |
| | State($C_3$) | Sectoral Carbon Emissions($E_9$) | Total carbon emissions from tourism traffic (10,000t) $I_{27}$ | Inverse | Zhang and Zhang (2020) [30] |
| | | | Total carbon emissions from tourism accommodation (10,000 t) $I_{28}$ | Inverse | |
| | | | Total carbon emissions from tourism activities (10,000 t) $I_{29}$ | Inverse | |
| | | Tourism Industry Carbon Emissions($E_{10}$) | Total carbon emissions from tourism (10,000t) $I_{30}$ | Inverse | Peng et al. (2022)[56]; Zhang and Zhang (2020) [30] |
| | | | Carbon emission intensity of tourism (t per10,000 RMB) $I_{31}$ | Inverse | |

(*Continued*)

**Table 1.** (Continued)

| Target | Criterions | Elements | Indicators | Character | References |
|---|---|---|---|---|---|
| | Impact($C_4$) | Economic Income($E_{11}$) | Ticket revenue (Billion RMB) $I_{32}$ | Positive | Zhang et al. (2022)[58] |
| | | | Ticket revenue from sea excursions (Billion RMB) $I_{33}$ | Positive | |
| | | | Comprehensive tourism revenues (Billion RMB) $I_{34}$ | Positive | |
| | | | Tourism revenue as a percentage of GDP (%) $I_{35}$ | Positive | |
| | | | Total retail sales of social consumer goods (10,000 RMB) $I_{36}$ | Positive | |
| | | | Average annual wage of employees (RMB) $I_{37}$ | Positive | |
| | | Ecological Environment ($E_{12}$) | Air quality conditions—average PM 2.5 concentration (μg/m3) $I_{38}$ | Inverse | Peng et al. (2022)[56]; Zhou et al. (2015)[47] |
| | | | Ambient air quality excellent rate (%) $I_{39}$ | Positive | |
| | | | County ambient air quality composite index $I_{40}$ | Positive | |
| | | | Sewage treatment rate (%) $I_{41}$ | Positive | |
| | | | Green space in built-up areas (acres) $I_{42}$ | Positive | |
| | Response ($C_5$) | Government Planning ($E_{13}$) | Energy conservation and environmental protection expenditure (10,000 RMB) $I_{43}$ | Positive | Chen et al. (2022)[59]; Hu et al. (2024)[57] |
| | | | Number of normative documents related to low carbon development (Unit) $I_{44}$ | Positive | |
| | | | Establishing low-carbon tourism demonstration areas $I_{45}$ | Positive | |
| | | | Proportion of low-carbon publicity (%) $I_{46}$ | Positive | |
| | | | Number of public bus operations (Unit) $I_{47}$ | Positive | |
| | | Corporate Practice($E_{14}$) | Clean energy usage (%) $I_{48}$ | Positive | Swangjang and Kornpiphat (2021)[53]; Cheng et al. (2013)[21] |
| | | | Energy-efficient equipment usage (%) $I_{49}$ | Positive | |

Abbreviations: GDP, Gross Domestic Product.

$$\lambda_{max} = \frac{1}{n}\sum_{i=1}^{n}\frac{A \cdot w_i^0}{w_i^0} \tag{8}$$

$$CI = \frac{\lambda_{max} - n}{n - 1} \tag{9}$$

$$CR = \frac{CI}{RI} \tag{10}$$

where $w_i^0$ is the relative weight of the element, $A$ is the judgment matrix, $a_{ij}$ is the weight vector, and $n$ is the number of indexes; $\lambda_{max}$ is the maximum eigenvalue, $CI$ is the consistency index, $RI$ is the average random consistency index, and $CR$ is the consistency ratio.

Based on the standardized data that were obtained by the nondimensionalization of the raw objective data, the entropy weight method was used to conduct a comprehensive evaluation of multiple indexes and calculate the index variability, information entropy, information entropy redundancy, index weights, and comprehensive score using Eqs (11)–(14), and finally the weight of each index was obtained in Table 2.

$$P_{ij} = \frac{X_{ij}}{\sum_{i=1}^{n} X_{ij}} \ (i = 1, 2, 3 \ldots, \text{n}; \ j = 1, 2, 3 \ldots, m) \tag{11}$$

$$e_j = -\frac{1}{\ln n} \sum_{i=1}^{n} P_{ij} \ln P_{ij} \tag{12}$$

$$g_j = 1 - e_j \tag{13}$$

$$w_j = \frac{g_j}{\sum_{j=1}^{m} g_j} \tag{14}$$

where $P_{ij}$ is the weight of $i$ to $j$, $e_j$ is the information entropy, $g_j$ is the information entropy redundancy, and $w_j$ is the index weight obtained by the entropy weight method; $n$ is the number of indexes and $m$ is the number of years.

Additionally, by substituting the AHP weights and the entropy weight method weights into Eq (15), the comprehensive weight $w$ of the indexes was obtained (Table 2). Using Eq (16), the final score of the low-carbon island tourism development level evaluation was calculated.

$$w = 0.5w_i + 0.5w_j \tag{15}$$

$$S_i = \sum_{j=1}^{m} wX_{ij} \tag{16}$$

## Data source

The raw data that were used in this study to measure the carbon emissions from tourism and to construct the evaluation system were obtained from the Zhong Guo Hai Dao Zhi, Statistical Yearbook of Yantai, Statistical Yearbook of Changdao Island, Changhai County Statistics Bulletin of National Economic and Social Development, Changdao Island Comprehensive Experimental Zone Statistics Bulletin of National Economic and Social Development, and other government websites. The government departments of the Changdao Island Comprehensive Experimental Zone Office, such as the Working Committee and Management Committee, the Transportation and Housing Construction Bureau, the Ministry of Publicity, the Ministry of Culture and Tourism, and the Bureau of Ecology and Environment, provided data support. The accommodation data, such as hotel reviews from the Ctrip website, were collected using the Bazhuayu web crawler. Additionally, some of the data were based on the results of previous studies at home and abroad or obtained from field research.

## Results

### Carbon emissions from tourism

According to Table 3, from 2016 to 2021, the total carbon emissions from tourism in Changdao Island showed a decreasing trend, which was related to the clean energy input and low-

**Table 2. Indicator weights.**

| Target (Weight) | Criterions (Weight) | Elements (Weight) | Indicators | Entropy method weight | AHP method weight | Combined weight |
|---|---|---|---|---|---|---|
| $O(1)$ | $C_1(0.1469)$ | $E_1(0.0450)$ | $I_1$ | 0.0209 | 0.0054 | 0.0131 |
| | | | $I_2$ | 0.0117 | 0.0043 | 0.0080 |
| | | | $I_3$ | 0.0165 | 0.0069 | 0.0117 |
| | | | $I_4$ | 0.0173 | 0.0069 | 0.0121 |
| | | $E_2(0.0415)$ | $I_5$ | 0.0130 | 0.0072 | 0.0101 |
| | | | $I_6$ | 0.0149 | 0.0072 | 0.0110 |
| | | | $I_7$ | 0.0123 | 0.0090 | 0.0107 |
| | | | $I_8$ | 0.0099 | 0.0094 | 0.0096 |
| | | $E_3(0.0420)$ | $I_9$ | 0.0099 | 0.0053 | 0.0076 |
| | | | $I_{10}$ | 0.0196 | 0.0108 | 0.0152 |
| | | | $I_{11}$ | 0.0099 | 0.0046 | 0.0073 |
| | | | $I_{12}$ | 0.0130 | 0.0108 | 0.0119 |
| | | $E_4(0.0184)$ | $I_{13}$ | 0.0312 | 0.0057 | 0.0185 |
| | $C_2(0.1917)$ | $E_5(0.0517)$ | $I_{14}$ | 0.0122 | 0.0156 | 0.0139 |
| | | | $I_{15}$ | 0.0122 | 0.0156 | 0.0139 |
| | | | $I_{16}$ | 0.0292 | 0.0184 | 0.0238 |
| | | $E_6(0.0626)$ | $I_{17}$ | 0.0176 | 0.0136 | 0.0156 |
| | | | $I_{18}$ | 0.0171 | 0.0136 | 0.0153 |
| | | | $I_{19}$ | 0.0176 | 0.0136 | 0.0156 |
| | | | $I_{20}$ | 0.0185 | 0.0136 | 0.0160 |
| | | $E_7(0.0259)$ | $I_{21}$ | 0.0124 | 0.0136 | 0.0130 |
| | | | $I_{22}$ | 0.0122 | 0.0136 | 0.0129 |
| | | $E_8(0.0515)$ | $I_{23}$ | 0.0122 | 0.0135 | 0.0129 |
| | | | $I_{24}$ | 0.0122 | 0.0135 | 0.0129 |
| | | | $I_{25}$ | 0.0122 | 0.0135 | 0.0129 |
| | | | $I_{26}$ | 0.0122 | 0.0135 | 0.0129 |
| | $C_3(0.1699)$ | $E_9(0.1037)$ | $I_{27}$ | 0.0342 | 0.0371 | 0.0356 |
| | | | $I_{28}$ | 0.0309 | 0.0371 | 0.0340 |
| | | | $I_{29}$ | 0.0312 | 0.0371 | 0.0341 |
| | | $E_{10}(0.0662)$ | $I_{30}$ | 0.0321 | 0.0371 | 0.0346 |
| | | | $I_{31}$ | 0.0261 | 0.0371 | 0.0316 |
| | $C_4(0.2828)$ | $E_{11}(0.1321)$ | $I_{32}$ | 0.0196 | 0.0269 | 0.0233 |
| | | | $I_{33}$ | 0.0141 | 0.0269 | 0.0205 |
| | | | $I_{34}$ | 0.0195 | 0.0269 | 0.0232 |
| | | | $I_{35}$ | 0.0112 | 0.0269 | 0.0191 |
| | | | $I_{36}$ | 0.0173 | 0.0269 | 0.0221 |
| | | | $I_{37}$ | 0.0222 | 0.0254 | 0.0238 |
| | | $E_{12}(0.1507)$ | $I_{38}$ | 0.0275 | 0.0290 | 0.0282 |
| | | | $I_{39}$ | 0.0190 | 0.0289 | 0.0240 |
| | | | $I_{40}$ | 0.0223 | 0.0290 | 0.0256 |
| | | | $I_{41}$ | 0.0592 | 0.0290 | 0.0441 |
| | | | $I_{42}$ | 0.0286 | 0.0290 | 0.0288 |
| | $C_5(0.2088)$ | $E_{13}(0.1448)$ | $I_{43}$ | 0.0168 | 0.0321 | 0.0245 |
| | | | $I_{44}$ | 0.0398 | 0.0319 | 0.0358 |
| | | | $I_{45}$ | 0.0238 | 0.0321 | 0.0279 |
| | | | $I_{46}$ | 0.0250 | 0.0319 | 0.0284 |
| | | | $I_{47}$ | 0.0220 | 0.0345 | 0.0282 |
| | | $E_{14}(0.0640)$ | $I_{48}$ | 0.0375 | 0.0343 | 0.0359 |
| | | | $I_{49}$ | 0.0220 | 0.0343 | 0.0281 |

**Table 3. Annual tourism carbon emissions by sector.**

| Item (Unit: Mt) / Year | 2016 | 2017 | 2018 | 2019 | 2020 | 2021 |
|---|---|---|---|---|---|---|
| **Transportation** | **0.6083** | **0.7549** | **0.7303** | **0.7450** | **0.3541** | **0.3936** |
| Ship and ferry | 0.4207 | 0.5280 | 0.4978 | 0.4995 | 0.2426 | 0.2664 |
| Fuel car | 0.1769 | 0.2138 | 0.2191 | 0.2314 | 0.1051 | 0.1198 |
| New energy car | 0.0017 | 0.0022 | 0.0022 | 0.0023 | 0.0011 | 0.0012 |
| Bus | 0.0090 | 0.0109 | 0.0112 | 0.0118 | 0.0054 | 0.0061 |
| Cycling and walking | 0 | 0 | 0 | 0 | 0 | 0 |
| **Accommodation** | **0.6140** | **0.6699** | **0.6628** | **0.6396** | **0.4549** | **0.5392** |
| Star-rated fisherman's family home hotel | 0.6081 | 0.6633 | 0.6564 | 0.6335 | 0.4506 | 0.5341 |
| Star-rated hotel bed occupancy | 0.0059 | 0.0066 | 0.0064 | 0.0061 | 0.0043 | 0.0051 |
| **Activities** | **0.0912** | **0.0996** | **0.0984** | **0.0950** | **0.0676** | **0.0802** |
| Sightseeing tour | 0.0186 | 0.0203 | 0.0201 | 0.0194 | 0.0138 | 0.0164 |
| Leisure holiday | 0.0676 | 0.0738 | 0.0730 | 0.0704 | 0.0501 | 0.0594 |
| Business trip | 0.0039 | 0.0043 | 0.0042 | 0.0041 | 0.0029 | 0.0034 |
| Other tourism purposes | 0.0011 | 0.0012 | 0.0012 | 0.0011 | 0.0008 | 0.0010 |
| Total | 1.3135 | 1.5244 | 1.4915 | 1.4796 | 0.8766 | 1.013 |

carbon transformation but there was also a close correlation with the downsizing of tourism. From 2016 to 2021, the annual carbon emissions from tourism were 1.3135, 1.5244, 1.4915, 1.4796, 0.8766, and 1.013 Mt, respectively.

Changdao Island is an off-shore island tourism destination and the central transportation mode that is used to reach the scenic spot is waterway transportation. Since 2018, to promote the construction of marine ecological civilization and to realize green travel on Changdao Island, there has been a ban on non-local vehicles from entering the island. Hence, the internal road transportation modes that were available for tourists on the island included leased cars, operating fuel cars, electric cars, buses, sightseeing vehicles, bicycles, and walking. In this study, the carbon emissions of the available internal transportation modes were as follows [35]: 0.07 kg/pkm for ships and ferries, 0.075 kg/pkm for fuel cars, 0.01 kg/pkm for electric cars, 0.018 kg/pkm for buses/sightseeing vehicles, and 0 kg/pkm for bicycles/walking. As the only way for tourists to enter the island, water transport was the main source of carbon emissions from tourism transportation, accounting for 67% -70%. During 2016–2019, with the change in the number of tourists, the tourist turnover for each transportation mode fluctuated slightly, and the carbon emissions from tourism transportation also showed a relatively stable trend. During the COVID-19 outbreak, tourism was drastically affected. Specifically, the number of tourists dropped significantly, the tourist turnover was reduced by about 50%, and the total carbon emissions from tourism transportation were reduced to 0.3541 Mt (2020) and 0.3936 Mt (2021) shown in Table 3.

In 1999, the "fishtainment" tourism model emerged in Changdao Island and Penglai, with fishermen's families in fishing villages as the main operation entity, and it integrated the experience of fishermen's folk customs, accommodation, catering, and leisure, with tourism and featured fishermen's lifestyle and fishing culture. Herein, the unit energy consumption of a star-rated fishtainment homestay was 40 MJ/bed-night [62], the unit energy consumption of star-rated hotels was 130 MJ/bed-night [40], and the conversion coefficient between energy consumption and carbon emissions was set to 43.2 g/MJ [7,63]. With the transition of tourists' tourism purpose to leisure and vacation, the accommodation and catering features of fishtainment become important, and fishtainment homestays became the major choice for overnight

tourists in Changdao Island. Thus, they are also the key source of carbon output from tourist accommodation. During 2016–2018, as the number of overnight tourists grew, carbon emissions rose accordingly. During 2018–2020, the carbon emissions from accommodation continuously decreased. During 2020–2021, the carbon emissions from accommodation increased significantly.

Based on previous studies [64] and the main tourism purpose of the Changdao Island tourists, the tourism activities and their unit energy consumption and carbon emissions were determined as follows: sightseeing (energy consumption: 8.5 MJ/capita; carbon emissions: 113.6 g/capita), leisure vacation (energy consumption: 26.58.5 MJ/capita; carbon emissions: 455.0 g/capita), business trip (energy consumption: 16 MJ/capita; carbon emissions: 214.2 g/capita), and other (energy consumption: 3.5 MJ/capita; carbon emissions: 46.9 g/capita). On this basis, the carbon release from the tourism activities of Changdao Island tourists during 2016–2021 were obtained in Table 3. The total carbon emissions from tourism activities were 0.0912, 0.0996, 0.0984, 0.0950, 0.0676, and 0.0802 Mt, respectively, during 2016–2021.

## Results of the evaluation of the low-carbon tourism development on the islands

In this section, according to the established low-carbon island tourism development level evaluation index system, the weight of each layer was obtained in Table 2, and the annual comprehensive score was obtained in Fig 2. Additionally, the annual variation in the low-carbon tourism development level, index weights, and score features were analyzed to explore the development pattern and key factors of low-carbon tourism development.

The "impact" was the most influential element of the island low-carbon tourism valuation with a weighted value of 0.2828. The "impact" indicator expressed the feedback role of the changes in the state of the island due to low-carbon tourism development on the ecological environment and economy. At the "impact" level, the ecological environment value was 0.1507 higher than that of the economic value (0.1321). This explains that the development of low-carbon tourism promoted regional air quality, sewage treatment, and green spaces. Apart from the environmental state, economic incomes, such as local employee earnings, ticket revenues, comprehensive tourism revenues, and total retail sales of social consumer goods, also increased due to low carbon development. Therefore, low-carbon tourism built up an industry

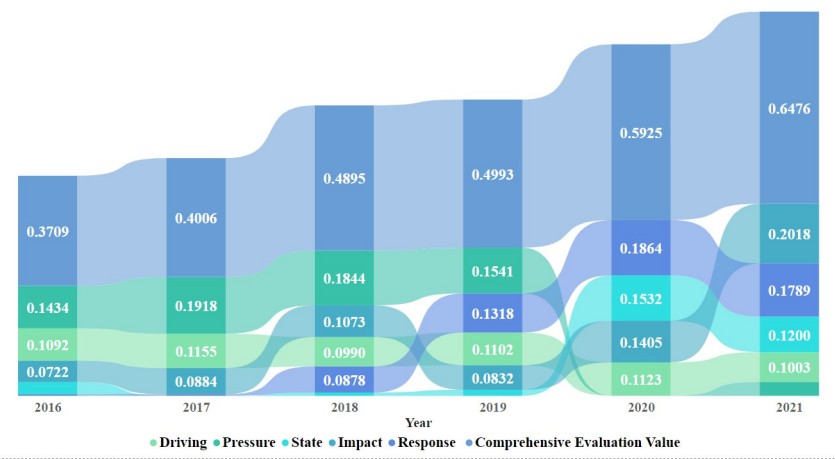

**Fig 2. Results of the evaluation of low carbon tourism on Changdao Island.**

chain system, with energy saving and consumption reduction helping tourists to have a better experience, which further enabled island tourism to occupy a larger tourism market.

The "response" also occupied a crucial position with a 0.2088 weighted value. It included the actions of people to prevent or mitigate negative impacts on the island environment. Furthermore, the "response" was divided into government planning and corporate practice. However, the weighted value of government planning was 0.1448, which was twice as high as that of corporate practice. This shows that the government plays a crucial role in low-carbon activities. This is especially true when the government establishes enough standard documents for low-carbon development in the tourism field, and this will increase the value of low-carbon tourism. Moreover, establishing low-carbon tourism demonstration areas, increasing low-carbon publicity, and increasing the number of public bus operations are all efficient methods to increase the value of low-carbon tourism. Additionally, clean energy and energy-efficient equipment usage by corporates also influenced the island tourism evaluation with weighted values of 0.0359 and 0.0281, respectively.

The next elements were "pressure" and "state". The "pressure" was the direct impact of tourism activities on the island's natural environment. It was the root cause of environmental problems at a weighting of 0.1917. The "pressure" statement for the Changdao area was calculated based on the scale of island tourism, transportation, accommodation, and tourism activities. These were the reasons for local environmental changes and problems. The "state" weighted value was 0.1699. This reflected the status of the carbon emissions from the tourism industry on Changdao Island. In contrast with the other indicators, all the indicators at the "state" level were negative. This is due to the negative effect of carbon emissions on the regional low-carbon development levels. All the "state" indicators had large weighted values for both the entropy weight and AHP methods. This is because carbon emissions are a key criterion for the island tourism environment.

The weight of the "driving forces" was 0.1469. They were the original deep-seated factors that affected the island's sustainable development. They usually refer to demographic, social, and economic changes that directly or indirectly affect economic changes. In this study, economic, social, resource, and geographical elements were chosen to identify the strength of the "driving forces". The geographical location was measured by the number of days with suitable temperatures for tourism. The geographical location's weighted value was 0.0184, which was smaller than the other indicators that were all larger than 0.04 at the "driving forces" level. The economic weighted value was 0.0450, which shows that economics could be one of the most important "driving forces".

Based on Table 1, the Entropy Weight and AHP Methods yielded almost identical weighting results. However, the Entropy Weight and AHP Method Weights of the economic indicators at the driving level were different. The Entropy Weight Method yielded a higher weight value than that which was obtained using the AHP method. The Entropy Weight Method obtains data that is based on the value of the data because the regional economic income data is a reflection of the regional development and contributes to the sustainable development of the region. Conversely, the weight of the economic income at the "impact" level that was obtained using the entropy weighting method was lower than that for the AHP method. The regional impact of low-carbon development features prominently in the economic income component, and therefore it received a high score for the AHP Method. In addition, the top five indicators with the highest weights were the sewage treatment rate at the impact level, the number of normative documents related to low carbon development and the rate of clean energy usage in enterprises at the response level, and the transportation carbon emissions and total carbon emissions at the state level. The remaining weighted indicators with high values were mainly from the State and Response levels. Thus, it can be generalized that the

government and enterprises' environmental protection actions have substantial value and significance for low carbon evaluation.

The low-carbon island tourism development evaluation index system comprised the objective layer (O), the criterion layer (C), the element layer (E), and the indicator layer (I). In terms of the criterion layer, the "impact" (C4) contributed the most to the low-carbon tourism development of Changdao Island, followed by the response (C5), pressure (C2), state (C3), and driving forces (C1). In terms of the element layer, the weights of the elements in descending order were as follows: the ecological environment (E12), government planning (E13), economic income (E11), carbon emissions from sub-sectors (E9), and carbon emissions from tourism (E10), and location (E4). Hence, the impact of tourism in Changdao Island on the ecological environment, local economy, and resident income was notable. The government support, planning, and management of low-carbon tourism can play an important role in promoting low-carbon tourism, and reducing the carbon emissions of the three core carbon source departments is the key to low-carbon tourism development. Moreover, the regional economic growth level will also become an important factor that drives the choice of tourists regarding the tourism destination. Then, in terms of the indicator layer, the weights of the elements in descending order were as follows: the sewage treatment rate (I41), rate of clean energy usage (I48), number of normative documents related to low-carbon development (I44), carbon emissions from tourism (I30), and total carbon emissions from tourism transportation (I27). These are in line with the main functions of environmental protection and the development direction of building an international zero-carbon demonstration zone in Changdao Island's comprehensive marine ecological civilization pilot zone.

## Conclusions and discussion

This study selected Changdao Island as the focus of the study and based on the data from 2016 to 2021, the carbon emissions from tourism in Changdao Island were calculated and the evaluation results were analyzed comprehensively. The main conclusions are specified below:

Firstly, the economic growth level of tourist destinations was an important factor driving tourists' choices and a major indirect reason for the increase in carbon emissions from tourism. When compared with factors, such as economy, society, and tourism resources, the driving force of the geographical location was the smallest. This is consistent with the study of energy carbon emissions in Yantai City, where Changdao Island is located, where economic factors, such as industrial structure and Gross Domestic Product growth, had the strongest driving force [65]. In addition, existing studies generally consider economy and society as the main driving factors, and tourism resources play an important role in the development of the tourism industry [34]. Therefore, in related research on tourism development, "resources" are also one of the main driving factors. However, few studies have considered geographical location factors in their analyses. Most of the world's major island tourism destinations are located in low-latitude areas, such as tropical and subtropical regions, where tourism development is less affected by the regional climate. However, Changdao Island is located in a temperate monsoon climate zone, characterized by distinct seasons and clear peak and off-peak tourism periods. Therefore, the location likely had an impact on the driving force of tourism development in this region.

Secondly, the "state" is critical for evaluating the low-carbon development level of tourism. In this study, "pressure" and "state" were closely related to the main project settings that generated carbon emissions within the tourism industry. The pressure resulting from the tourism scale on island destinations cannot be underestimated; therefore, the ecological carrying capacity of islands remains a focus of research on island tourism [66]. When measuring the

tourism carrying capacity in Chinese island cities, Changdao Island performed the worst [67]. Therefore, for Changdao Island, we would caution against the development of large-scale tourism, especially low-carbon tourism transportation options for stakeholders. This aligns with the findings of other scholars [31,65].

Thirdly, the development of low-carbon tourism on islands significantly impacts both the destination economy and its ecological environment. Environmental and economic impacts are critical criteria in research on green and sustainable development [68]. Low-carbon tourism can increase local tourism revenue and consumption levels, promote sustained economic growth, and enhance investment and human resources, thereby creating a driving force for sustainable development. Furthermore, in island tourism destinations, natural resources are fundamental for the growth of the tourism industry. The low-carbon development of tourism supports the sustainable use of natural resources and provides an eternal driving force for development. In addition, for residents of tourist destinations, the increase in tourism income enhances their economic benefits and their awareness of environmental protection. Consequently, for zero carbon construction on island tourism destinations, such as Chang Island, relevant indicators should be carefully considered.

Fourthly, governments play a crucial role in guiding the development of low-carbon tourism, and stakeholders also play a role in implementing policies. In the research on the sustainable development of the tourism industry, it has also been concluded that the government and stakeholders must work together [69]. Currently, in promoting the development of low-carbon tourism on Changdao Island, the government has formulated standard documents to guide the behavior of stakeholders, established low-carbon tourism demonstration zones, increased low-carbon publicity efforts, and increased the number of public transportation operations. Fourthly, stakeholders can also practice low-carbon development through investment in energy-saving and carbon-reduction technologies.

## Managerial implications

Low-carbon tourism is the basic requirement and correct direction for island tourism development. The development of tourism with a low carbon focus will build a protection network for the green and sustainable development of the island ecosystem and it will provide a driving force for the transformation of the island and fishing villages from a traditional development model to a modern development model. Moreover, it can revitalize local economic development and improve the incomes, living standards, and quality of life of the residents of the islands. Although the low-carbon tourism development level in Changdao Island is gradually increasing, there is still a large potential for improvement. Currently, the local government is actively promoting the construction of Changdao Island International Zero Carbon Island and has proposed the establishment of a zero-carbon tourism destination. Considering the dynamic nature of low-carbon tourism development, there is a pressing need to create a comprehensive and systematic indicator system and effective evaluation methods for ongoing monitoring and evaluation. Therefore, we hope that the evaluation index system for low-carbon tourism development that was established in this study will serve as a theoretical framework for monitoring and evaluating zero-carbon tourism.

In light of the three sub-sectors of tourism, the carbon emissions from tourism in Changdao Island were mainly due to tourism transportation and tourist accommodation. When compared with the carbon emissions from other tourism destinations, the carbon emissions from tourism transportation in Changdao Island made up a lower proportion, which was mainly because tourists can only enter the island by ship, which is a central transportation mode with low unit carbon emissions. Additionally, since the end of 2018, private tourist cars

were banned from entering Changdao Island, so tourists can only enter the island by ship and use electric cars, shared bicycles, electric bicycles, public electric buses, taxis, walking, and other available vehicles for internal transportation. This promotes carbon reduction. However, since the implementation of the vehicle policy in 2018, the tourism person-time decreased for the first time in 2019 after a continuous slow increase. The ban on private cars has had certain negative impacts on the number of tourists choosing Changdao Island as a tourism destination and constrains tourism development on Changdao Island. Hence, learning from the experience of mass tourism transport planning in coastal areas, island-hopping routes should be built based on factors such as attractions, tourism facilities, and geographical conditions, and convenient tourism services should be introduced, such as tourist sightseeing buses and self-drive electric car hire and charging services. The carbon emissions from tourist accommodation made up a relatively higher proportion. The accommodations are limited by the high costs of clean energy, and fishtainment homestay operators have a low rate of clean energy usage. In light of this, relevant government departments should further encourage and support the use of clean energy in the tourism accommodation sector by increasing clean energy inputs and providing financial incentives while promoting the special fishtainment homestay construction project on Changdao Island. Furthermore, environmentally friendly advertisements, points rewards, or gifts can increase visitors' awareness of conserving resources and reducing the use of disposable items.

## Limitations and future research directions

The main purpose of this study was to develop a causal effect framework based on the DPSIR model to comprehensively and systematically evaluate the development trends of low-carbon tourism on islands. To ensure the comprehensiveness and accuracy of the evaluation indicators, we constructed the overall framework using relevant data from the Changdao Island Marine Ecological Civilization Comprehensive Experimental Zone. However, since this study involves county-level indicators, the types of indicators may evolve due to changes in the statistical organization and personnel. Thus, the application of this framework for long-term and regular evaluations in the region has certain limitations. Nevertheless, this study can serve as a foundation for establishing a zero-carbon tourism destination on Changdao Island and provide a theoretical basis for long-term and regular statistical indicators. In the future, we also plan to evaluate tourism destinations in China and other islands worldwide, and through repeated adjustments and optimizations, build a more widely applicable evaluation model.

In addition, in light of the global carbon reduction goals and China's dual carbon goals, this study focused on measuring the carbon emissions of island tourism destinations and evaluating the level of low-carbon tourism development. The aim was to identify key factors that promote zero-carbon initiatives and to assist policymakers and tourism planners in providing targeted policy guidance and project planning. However, this study did not investigate the interrelationships among the subsystems and elements in the DPSIR model. Future research should develop comprehensive evaluation models, such as the DPSIR model, structural equation model, data envelopment analysis, and coupled co-scheduling model, to further investigate the degree of mutual influence, efficiency evaluation, synergistic effects, and trends within these frameworks.

## Supporting information

**S1 Data. Data of Island low carbon tourism development evaluation index system.**
(DOCX)

**S2 Data. AHP data.**
(ZIP)

## Acknowledgments

The authors would like to thank the reviewers and editors, as well as others who helped with the manuscript and whose suggestions greatly improved the manuscript.

## Author Contributions

**Conceptualization:** Mengsha Wang.

**Data curation:** Mengsha Wang, Jiayu Zuo.

**Funding acquisition:** Mengsha Wang.

**Investigation:** Mengsha Wang.

**Methodology:** Jiayu Zuo.

**Validation:** Jiayu Zuo.

**Visualization:** Jiayu Zuo.

**Writing – original draft:** Mengsha Wang.

**Writing – review & editing:** Jiayu Zuo.

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
