## [Decision Letter · Decision Letter 0]

27 Aug 2024

PONE-D-24-10283Measurement and evaluation of low-carbon tourism development on islands: A case study in Changdao, ChinaPLOS ONE

Dear Dr. ­ZUO,

Thank you for submitting your manuscript to PLOS ONE. After careful consideration, we feel that it has merit but does not fully meet PLOS ONE’s publication criteria as it currently stands. Therefore, we invite you to submit a revised version of the manuscript that addresses the points raised during the review process.

We look forward to receiving your revised manuscript.

Kind regards,

Vincenzo Basile, PhD

Academic Editor

PLOS ONE

 [This study was financially supported by Shandong Technology and Business University 's Doctoral Introduction Start-up Fund Project--Research on Economic Benefits and Development Countermeasures of Carbon Sink Fisheries in the Blue Economic Zone of Shandong Peninsula (Fund No. BS2021144).].  

4. We notice that your supplementary figures are uploaded with the file type 'Figure'. Please amend the file type to 'Supporting Information'. Please ensure that each Supporting Information file has a legend listed in the manuscript after the references list.

Additional Editor Comments (if provided):

Reviewers' comments:

Reviewer's Responses to Questions

**Comments to the Author**

1. Is the manuscript technically sound, and do the data support the conclusions?

Reviewer #1: Yes

Reviewer #2: Partly

2. Has the statistical analysis been performed appropriately and rigorously? 

Reviewer #1: Yes

Reviewer #2: Yes

3. Have the authors made all data underlying the findings in their manuscript fully available?

Reviewer #1: Yes

Reviewer #2: Yes

4. Is the manuscript presented in an intelligible fashion and written in standard English?

Reviewer #1: Yes

Reviewer #2: Yes

5. Review Comments to the Author

Reviewer #1: Strengthen the literature review by incorporating more comprehensive references related to low-carbon tourism, island tourism development, and sustainable tourism practices.

Provide a more detailed description of the methodology used to construct the evaluation model for low-carbon tourism.

Discuss the significance of each indicator and its contribution to the overall assessment of low-carbon tourism development on Changdao Island.

Compare the findings with similar studies or benchmarks to contextualize the results and identify areas for improvement.

Clear the implications of the findings for policymakers, stakeholders, and researchers involved in promoting sustainable tourism.

Provide specific recommendations based on the results to guide future decision-making and action in advancing low-carbon tourism initiatives on islands.

Discuss the potential long-term effects of implementing low-carbon tourism measures on Changdao Island.

Add these relevant references

- Analysis of factors and strategies for the implementation of sustainable tourism in a green economic structure in China

- Exploring the Complex Nexus between Sustainable Development and Green Tourism through Advanced GMM Analysis

- Assessing impact investing for green infrastructure development in low-carbon transition and sustainable development in China

Address sustainability concerns such as the resilience of ecological systems, the socio-economic impact on local communities, and the scalability of sustainable practices.

Incorporate discussions on adaptive management strategies to address uncertainties and promote ongoing improvement.

Reviewer #2: The manuscript needs to be corrected before the publication procedure. it does suffer from several formal inconsistencies that need to be addressed. Additionally, there are minor suggestions regarding the content, but the main focus should be on revising the formal structure of the paper. Please also check for any typing errors such as missing or extra blank spaces and lines, correct use of capital letters, and ensure that there is no plagiarism detected by the similarity score, it’s about 18 %.,

6. PLOS authors have the option to publish the peer review history of their article (what does this mean?). If published, this will include your full peer review and any attached files.

Reviewer #1: No

Reviewer #2: **Yes: **Musallam R. Al-Rawahneh

---

## [Author Response · Author response to Decision Letter 0]

27 Sep 2024

We are very grateful for the reviewer’s thoughtful feedback and suggestions, which greatly contributed to the improvement of our research. Their input has not only enhanced the clarity of our manuscript but also added depth to our discussion.

---

## [Decision Letter · Decision Letter 1]

8 Oct 2024

Measurement and evaluation of low-carbon tourism development on islands: a case study in Changdao, China

PONE-D-24-10283R1

Dear Dr. Jiayu | 재무금융전공 | 한양대(서울) ­ZUO,

We’re pleased to inform you that your manuscript has been judged scientifically suitable for publication and will be formally accepted for publication once it meets all outstanding technical requirements.

Kind regards,

Vincenzo Basile, PhD

Academic Editor

PLOS ONE

Additional Editor Comments (optional):

Reviewers' comments:

Reviewer's Responses to Questions

**Comments to the Author**

1. If the authors have adequately addressed your comments raised in a previous round of review and you feel that this manuscript is now acceptable for publication, you may indicate that here to bypass the “Comments to the Author” section, enter your conflict of interest statement in the “Confidential to Editor” section, and submit your "Accept" recommendation.

Reviewer #1: All comments have been addressed

Reviewer #2: All comments have been addressed

2. Is the manuscript technically sound, and do the data support the conclusions?

Reviewer #1: Yes

Reviewer #2: Yes

3. Has the statistical analysis been performed appropriately and rigorously? 

Reviewer #1: Yes

Reviewer #2: Yes

4. Have the authors made all data underlying the findings in their manuscript fully available?

Reviewer #1: No

Reviewer #2: Yes

5. Is the manuscript presented in an intelligible fashion and written in standard English?

Reviewer #1: Yes

Reviewer #2: Yes

6. Review Comments to the Author

Reviewer #1: (No Response)

Reviewer #2: (No Response)

7. PLOS authors have the option to publish the peer review history of their article (what does this mean?). If published, this will include your full peer review and any attached files.

Reviewer #1: No

Reviewer #2: **Yes: **Musallam R. Al-Rawahneh

---

## [Editor Report · Acceptance letter]

2 Jan 2025

PONE-D-24-10283R1 

PLOS ONE

Dear Dr. Zuo, 

I'm pleased to inform you that your manuscript has been deemed suitable for publication in PLOS ONE. Congratulations! Your manuscript is now being handed over to our production team.

Kind regards, 

on behalf of

Dr. Vincenzo Basile 

Academic Editor

PLOS ONE